# Toward Better PAC-Bayes Bounds for Uniformly Stable Algorithms

**Sijia Zhou**[1]    **Yunwen Lei**[2]*    **Ata Kabán**[1]

[1]School of Computer Science, University of Birmingham, Birmingham B15 2TT, United Kingdom
[2] Department of Mathematics, The University of Hong Kong, Pokfulam, Hong Kong, China
`sxz115@student.bham.ac.uk`   `leiyw@hku.hk`   `a.kaban@bham.ac.uk`

## Abstract

We give sharper bounds for uniformly stable randomized algorithms in a PAC-Bayesian framework, which improve the existing results by up to a factor of $\sqrt{n}$ (ignoring a log factor), where $n$ is the sample size. The key idea is to bound the moment generating function of the generalization gap using concentration of weakly dependent random variables due to Bousquet et al (2020). We introduce an assumption of sub-exponential stability parameter, which allows a general treatment that we instantiate in two applications: stochastic gradient descent and randomized coordinate descent. Our results eliminate the requirement of strong convexity from previous results, and hold for non-smooth convex problems.

## 1 Introduction

Stochastic optimization methods are the workhorses of many modern machine learning problems. A lot of progress has been made in reducing optimization errors with less training time in stochastic settings [6, 21, 41, 46, 64]. One such method is stochastic gradient descent (SGD), which builds a stochastic gradient estimator iteratively based on a randomly sampled example to approximate the gradient for the next iteration. SGD is appealing for large-scale data analysis due to its cheap computational cost, simplicity and efficiency.

Obtaining models that generalize well is the goal in machine learning and we undertake a theoretical analysis of this problem. The generalization behavior of a model can be quantified by the excess risk, which decomposes into two components: the optimization error and the generalization error (or generalization gap). In this paper, we focus on the analysis of the generalization error of stochastic optimization methods. We combine two useful approaches to bound the generalization error, that is algorithmic stability [7, 16] and PAC-Bayes bounds [9, 26, 34, 53]. Our hope is to retain the benefits of both approaches to derive powerful generalization bounds to better understand the behavior of stochastic optimization methods.

Seminal work by Bousquet et al. [7] provided generalization bounds for uniformly stable algorithms in the deterministic case. Sharper generalization bounds were obtained via a moment bound on the generalization gap and a concentration inequality of weakly dependent random variables [8, 18]. Extending stability-based bounds to randomized algorithms is a challenge. In [16], where the stability-based bounds for the randomized algorithms were considered, the results hold for fixed distributions. To give bounds uniformly for all distributions, we need to turn to PAC-Bayes analysis. A recent work analyzed stable algorithms in the PAC-Bayes framework and derived generalization bounds that hold for any distribution [32]. However, a comparison between [8] and [32] reveals that the error convergence rate in the randomized case [32] is much slower than that for the deterministic case [8]. In more detail, it was shown that $\beta$-uniformly stable ($\beta$ is decreasing w.r.t. $n$, where $n$ is the sample

---

*Corresponding author

size) and deterministic algorithms would imply generalization bounds of order $\widetilde{O}(1/\sqrt{n}+\beta)$ [8, 18][2], while PAC-Bayes bounds of order $O(1/\sqrt{n}+\sqrt{n}\beta)$ were developed for randomized algorithms [32]. In [32], the randomness comes from the sampling of hyperparameters such as the index of examples chosen in SGD, and the PAC-Bayes bounds hold for any posterior distribution which may depend on the dataset. It is clear that the PAC-Bayes bounds can be slower than those in [8, 18] by a factor of $\sqrt{n}$, which motivates a natural question: can we develop PAC-Bayes bounds for randomized algorithms which match the rate of deterministic algorithms?

This paper explores the above question. We provide sharper PAC-Bayes bounds that hold for uniformly stable randomized algorithms. We adapt a moment bound [8] previously used for stable algorithms in deterministic cases and extend it to the PAC-Bayesian framework, to give bounds that hold for randomized predictors. The PAC-Bayes framework is based on the work of [32]. However, we take a different analysis strategy to control the change in hyperparameters, which is based on an assumption of sub-exponential stability parameter. This general assumption allows us to bound the moment generating function (MGF) of the generalization gap within a high probability domain where our assumption holds (see Appendix A.2). Furthermore, we prove that this assumption holds for both SGD and randomized coordinate descent (RCD). We then illustrate the advantage of our results over existing bounds.

Regarding the convergence rate, our main result improves on the existing PAC-Bayes bounds [32] by a factor of $\sqrt{n}$ (ignoring a log factor). This improvement holds under weaker conditions under which convergence is not guaranteed in [32]. Our primary technical tool is a moment bound, which we extend to randomized learning algorithms in the PAC-Bayes framework. This allows our bounds to hold for all possible posteriors, not just fixed ones [16] or deterministic algorithms [8, 18]. We need to introduce novel techniques to handle the randomness of the hyperparameter, which is a challenge in the PAC-Bayes analysis (details will be given in Section 3.2).

Regarding assumptions, our result holds without the requirement of hyperparameter stability in previous work [32]. Instead, we introduce a new assumption on the sub-exponential behavior of uniform stability by viewing the uniform stability as a function of the random hyperparameter. Interestingly, it suffices to study the sub-exponential behavior of uniform stability under the prior distribution, which makes the sub-exponential assumption easy to check.

We show that the uniform stability of SGD and RCD have a mixture of sub-Gaussian and sub-exponential tails that satisfy this assumption. Thus, we remove the strong convexity assumption in the existing analysis of SGD [32] and extend the result to non-smooth problems. Our result also applies to RCD, where the randomness arises from the selected coordinates.

The remainder of the paper is organized as follows. We discuss the basics on PAC-Bayes and stability analysis in Section 2. We present our main results in Section 3, and apply it to SGD and RCD in Section 4. We survey the related work on stability and PAC-Bayesian analysis in Section 5, and conclude the paper in Section 6.

## 2 Preliminaries

We first introduce some notations. Let $\mathcal{X}$ and $\mathcal{Y}$ denote the input and output space respectively and let $\mathcal{Z} = \mathcal{X} \times \mathcal{Y}$. We are given a set of training examples of size $n$, $S = \{z_1 = (x_1, y_1), \ldots, z_n = (x_n, y_n)\}$, drawn independently and identically distributed (i.i.d.) from an unknown distribution $\mathcal{D}$ on $\mathcal{Z}$. We hope to learn a predictor from a class of hypotheses $\mathcal{H}$ to predict unseen new data drawn from $\mathcal{D}$. Let $\mathcal{W}$ denote the weight space, $\mathcal{W} \subseteq \mathbb{R}^d$, and $\Theta$ denote a hyperparameter space. A deterministic learning algorithm $A : \mathcal{Z}^n \times \Theta \to \mathcal{H}$ maps the training examples to a hypothesis $h_{\mathbf{w}} \in \mathcal{H}$ determined by $\mathbf{w} \in \mathcal{W}$. The quality of a hypothesis is measured by a loss function, $\ell : \mathcal{H} \times \mathcal{Z} \to [0, M]$.

The goal of any learning algorithm is to produce a predictor that generalizes well. That means that the learned predictor applied on previously unseen input data from the marginal of $\mathcal{D}$ should have a small expected loss. For any $\theta \in \Theta$, the risk of a predictor returned by the algorithm $A$ is a random variable as a function of $S$, defined as

$$R(A(S; \theta)) = \mathbb{E}_{z \sim \mathcal{D}}[\ell(A(S; \theta), z)]. \tag{2.1}$$

---

[2]We use $\widetilde{O}$ to hide poly-logarithmic factors.

Since $\mathcal{D}$ is unknown, we have no access to the true risk. Instead, the empirical risk is often used as an approximation

$$R_S(A(S;\theta)) = \frac{1}{n}\sum_{i=1}^{n}\ell(A(S;\theta), z_i). \tag{2.2}$$

We are interested in how well the empirical risk can estimate the risk. The difference between them is the generalization gap $G(S;\theta) \triangleq R(A(S;\theta)) - R_S(A(S;\theta))$. We can upper bound the risk by bounding this difference.

## 2.1 PAC-Bayes basics

We are interested in randomized algorithms, such as stochastic gradient descent (SGD) and randomized coordinate descent (RCD). In such algorithms, there is an in-built random sampling mechanism that we can think of as a random hyperparameter. In other words, a randomized algorithm may be viewed as a deterministic algorithm with hyperparameters $\theta$ that follow a distribution. Therefore, for the analysis that follows, we will explicitly define distributions on $\Theta$. There has been a lot of interest in SGD because it is often used to train a model and its randomness comes from independent sampling of training instances to estimate gradient directions. In this case, the hyperparameters $\theta \in \Theta$ form a random sequence $\theta = (\theta_1, ..., \theta_T)$, where every $\theta_t \in [n]$, for $t \in [T]$, is an i.i.d. index of a training point from $S$. Here $[n]$ denotes $\{1, \ldots, n\}$.

Any distribution over the hyperparameter space $\Theta$ induces a distribution over the predictors (or weights since we often consider parametric models). In the PAC-Bayes methodology, the common approach is to define a prior distribution on the domain of the weights [13, 22, 34, 53]. However, in the context of randomized algorithms of the kind mentioned above it is more natural to exploit the randomness already present through a distribution on the domain of the hyperparameters.

Hence, we will conduct PAC-Bayesian analysis by focusing on the randomness of the hyperparameters. Let $\mathbb{P}$ be any prior distribution on the hyperparameter domain $\Theta$, chosen before seeing the training data. Given a randomized algorithm, we will only need to prove a condition for the prior $\mathbb{P}$ on $\Theta$, and in return our generalization guarantees will hold with high probability for *all* posteriors $\mathbb{Q}$ on $\Theta$. In particular, we can assume a simple uniform distribution as a prior for SGD, i.e. SGD with uniform sampling, and derive bounds for data-dependent non-uniform sampling [49, 56, 66]. For example, SGD with importance sampling [66] shows better results than uniform sampling, and there are efforts to learn good sampling distributions [32] in a line of research towards self-certified learning algorithms [13, 44].

The quality of a predictor learned by an algorithm $A$ from the training sample $S$, is defined as the expected risk w.r.t the PAC-Bayes posterior $\mathbb{Q}$ as

$$R(S, \mathbb{Q}) = \mathop{\mathbb{E}}_{\theta \sim \mathbb{Q}}[R(A(S;\theta))]. \tag{2.3}$$

Its empirical counterpart, the expected empirical risk w.r.t. $\mathbb{Q}$ is

$$R_S(S, \mathbb{Q}) = \mathop{\mathbb{E}}_{\theta \sim \mathbb{Q}}[R_S(A(S;\theta))]. \tag{2.4}$$

In special cases this expectation can be computed analytically; most often it is approximated either by Monte Carlo sampling or by analytical upper bounds.

PAC-Bayes bounds aim to estimate $R(S, \mathbb{Q})$ in terms of $R_S(S, \mathbb{Q})$ and the divergence between $\mathbb{P}$ and $\mathbb{Q}$. A key ingredient that PAC-Bayes bounds rest upon is a change of measure inequality, also known as variational formula. For completeness this is given in Appendix (Lemma A.3).

## 2.2 Algorithmic stability basics

Another relatively recent framework for generalization analysis is based on the concept of algorithmic stability [7]. The key concept in this framework quantifies how sensitive a learning algorithm is to small perturbations of the training data. There are several notions of algorithmic stability, and the one we use in our work is the following uniform stability [7]. We denote $S \sim S'$ if they are neighboring datasets, i.e., $S$ and $S'$ differ by at most a single example.

**Definition 1** (Uniform Stability). For any $\theta$, an algorithm $A : S \mapsto A(S; \theta)$ is $\beta_\theta$-uniformly stable w.r.t. a loss function $\ell$ if $\forall S \sim S' \in \mathcal{Z}^n, \forall z \in \mathcal{Z}$,

$$|\ell(A(S; \theta), z) - \ell(A(S'; \theta), z)| \leq \beta_\theta. \tag{2.5}$$

Recall that $A$ is a randomized algorithm whose randomness is exclusively due to a random draw of $\theta$. Hence, given a fixed instance of $\theta$, the algorithm $A$ becomes a deterministic algorithm. This simple observation helps us reduce the problem from randomized learning to deterministic learning. Based on this observation, the next section presents sharper generalization bounds for randomized algorithms in comparison to previous results [32].

## 3 Main Results

First we introduce a sub-exponential assumption on the stability parameter. This will allow us to make some key innovations: 1) We will be able to remove one of the assumptions required by [32, Theorem 2] (hyperparameter stability) and only require uniform stability. 2) we will state our result in more general terms, and instantiate it to specific algorithms such as SGD and RCD.

**Assumption 1** (Sub-exponential stability). Let $\mathbb{P}$ be a fixed probability distribution on $\Theta = \prod_{t=1}^{T} \Theta_t$. We say that a randomized algorithm with random hyperparameters $\theta \sim \mathbb{P}$ satisfies sub-exponential stability if, for any fixed instance of $\theta$ it satisfies $\beta_\theta$-uniform stability w.r.t. a loss function $\ell$, and there exists $c \in \mathbb{R}$ such that for any $\delta \in (0, 1/n]$, with probability at least $1 - \delta$ over draws of $\theta \sim \mathbb{P}$:

$$\beta_\theta \leq \mathbb{E}_{\theta \sim \mathbb{P}}[\beta_\theta] + c \log(1/\delta). \tag{3.1}$$

In other words, this assumption says that the deviation of $\beta_\theta$ from its mean roughly has a sub-exponential tail. We will show that $c$ is dominated by $\mathbb{E}_{\theta \sim \mathbb{P}}[\beta_\theta]$ in typical stochastic optimization algorithms such as SGD and RCD.

Note that in the above assumption, the probability is made w.r.t. $\theta \sim \mathbb{P}$, which is a prior distribution independent of $S$. We can choose $\mathbb{P}$ to be a uniform distribution. In this case, we will show that SGD satisfies this assumption under mild conditions on the loss functions with very small $\mathbb{E}_{\theta \sim \mathbb{P}}[\beta_\theta]$ and $c$.

Recall that for any fixed instance of $\theta$, the trained model of the learning algorithm $A$ on data set $S$ with hyperparameters $\theta$ is a deterministic predictor. Our proof strategy for our main result below is to first fix $\theta$, and apply a recent technique of obtaining sharp bounds [8] to the resulting deterministic algorithm. After that, we deal with the randomness of $\theta$.

In Theorem 1 we give our main result. This bound is useful when we have small $\mathbb{E}_{\theta \sim \mathbb{P}}[\beta_\theta]$ and $c$. We will see two examples in Section 4. We denote $B \gtrsim B'$ if there exists a universal constant $c_1 > 0$ such that $B \geq c_1 B'$. We use $B \lesssim B'$ if there exists a universal constant $c_2 > 0$ such that $B \leq c_2 B'$. We use $B \asymp B'$ if $B \lesssim B' \lesssim B$. The proof is given in Appendix A.2.

**Theorem 1** (Generalization of sub-exponentially stable randomized algorithms). *Consider a learning algorithm $A(S; \theta)$ that satisfies Assumption 1 w.r.t. $\mathbb{P}$ and $c \lesssim \mathbb{E}_{\theta \sim \mathbb{P}}[\beta_\theta]$. Assume $\ell(A(S; \theta), z) \in [0, M]$. Then for any $\delta_1 \in (0, 1)$ the following inequality holds with probability at least $1 - \delta_1$ uniformly for all $\mathbb{Q}$*

$$\mathbb{E}_{\theta \sim \mathbb{Q}}\big[R(A(S; \theta)) - R_S(A(S; \theta))\big] \lesssim \Big(D_{\mathrm{KL}}(\mathbb{Q}\|\mathbb{P}) + \log(1/\delta_1)\Big) \max\Big\{\mathbb{E}_{\theta \sim \mathbb{P}}[\beta_\theta] \log^2 n, \frac{M}{\sqrt{n}}\Big\},$$

*where $D_{\mathrm{KL}}(\mathbb{Q}\|\mathbb{P})$ means the KL divergence between $\mathbb{P}$ and $\mathbb{Q}$, i.e., $D_{\mathrm{KL}}(\mathbb{Q}\|\mathbb{P}) = \mathbb{E}_{\theta \sim \mathbb{Q}}\Big[\log \frac{\mathbb{Q}(\theta)}{\mathbb{P}(\theta)}\Big]$.*

To apply Theorem 1, we only need to check the sub-exponential assumption w.r.t. the prior $\mathbb{P}$. Then, we use the PAC-Bayesian analysis to transfer this stability assumption w.r.t. the specific $\mathbb{P}$ to a bound holding for all posterior distributions.

### 3.1 Comparison

Next, we compare our result (Theorem 1) with the previous generalization bounds on randomized algorithms due to [32] and observe the advantages that our approach offers.

To blend the stability into the PAC-Bayes framework, where hyperparameters $\theta$ follow a distribution, [32] defined the following new hyperparameter stability assumption for learning algorithms (Definition 2) and obtained the stability-based PAC-Bayes bound given in Theorem 2.

**Definition 2** (Hyperparameter Stability). A learning algorithm $A$ has uniform hyperparameter stability $\beta_\Theta$ w.r.t. the loss function $\ell$, if

$$\sup_{S \in \mathcal{Z}^n} \sup_{z \in \mathcal{Z}} \sup_{\theta, \theta' \in \Theta : D_H(\theta, \theta') = 1} |\ell(A(S; \theta), z) - \ell(A(S; \theta'), z)| \leq \beta_\Theta, \tag{3.2}$$

where $D_H(\mathbf{v}, \mathbf{v}') \triangleq \sum_{i=1}^{|\mathbf{v}|} \mathbb{I}[v_i \neq v_i']$ is the Hamming distance and $\mathbb{I}[\cdot]$ denotes the indicator function, i.e., $\mathbb{I}[E] = 1$ if the event $E$ holds and 0 otherwise.

Observe that, uniform stability (Definition 1) concerns the stability of an algorithm with respect to a change in the training set. In contrast, the above Definition 2 requires stability w.r.t. a change in the hyperparameters. Moreover, the approach in [32] requires stability w.r.t. both the loss function and the hyperparameters to derive the following PAC-Bayes bound.

**Theorem 2** (Theorem 2 of [32]). *Let $A$ be a randomized learning algorithm and $\ell(A(S; \theta), z) \in [0, M]$. Assume $A$ is $\beta_\theta$-uniformly stable w.r.t. loss functions, and $\beta_\Theta$-uniformly stable w.r.t. hyperparamters. Consider the prior $\mathbb{P}$ as a fixed probability distribution defined on $\Theta = \prod_{t=1}^{T} \Theta_t$. Then for any $n \geq 1$, $T \geq 1$, and $\delta \in (0, 1)$, the following inequality holds with probability at least $1 - \delta$ over draws of a data set, $S \sim \mathbb{D}^n$, for every posterior $\mathbb{Q}$ on $\Theta$*

$$\mathbb{E}_{\theta \sim \mathbb{Q}}\big[R(A(S; \theta)) - R_S(A(S; \theta))\big] \lesssim \sqrt{\Big(D_{\mathrm{KL}}(\mathbb{Q}\|\mathbb{P}) + \log(1/\delta)\Big)\Big((M + n\mathbb{E}_{\theta \sim \mathbb{P}}[\beta_\theta])^2/n + T\beta_\Theta^2\Big)}.$$

We now compare our bound (Theorem 1) with the bound above (Theorem 2).

In terms of the rate, our bound in Theorem 1 improves the previous result of Theorem 2 by up to a factor of $\sqrt{n}$ (up to poly-logarithmic factors when the divergence is polylogarithmic in $n$ [32]). To see this, consider the case where in both Theorem 1 and Theorem 2, the term that contains the stability parameter dominates over the KL term. Then, in Theorem 1, for $\beta_\theta$-uniformly stable randomized algorithms, we have

$$\mathbb{E}_{\theta \sim \mathbb{Q}}\big[R(A(S; \theta)) - R_S(A(S; \theta))\big] = \max\left\{\widetilde{O}(\mathbb{E}_{\theta \sim \mathbb{P}}[\beta_\theta]), \widetilde{O}(n^{-\frac{1}{2}})\right\} D_{\mathrm{KL}}(\mathbb{Q}\|\mathbb{P}).$$

In contrast, Theorem 2 in [32] gives

$$\mathbb{E}_{\theta \sim \mathbb{Q}}\big[R(A(S; \theta)) - R_S(A(S; \theta))\big] = \widetilde{O}\big(\sqrt{n}\mathbb{E}_{\theta \sim \mathbb{P}}[\beta_\theta] + \sqrt{T}\beta_\Theta\big) D_{\mathrm{KL}}^{\frac{1}{2}}(\mathbb{Q}\|\mathbb{P}). \tag{3.3}$$

It is clear that, in the case when $\widetilde{O}(\sqrt{n}\mathbb{E}_{\theta \sim \mathbb{P}}[\beta_\theta])$ dominates the KL divergence then having replaced it with $\widetilde{O}(\mathbb{E}_{\theta \sim \mathbb{P}}[\beta_\theta])$ we improved (3.3) by up to a factor of $\sqrt{n}$ (ignoring a log term). The prior $\mathbb{P}$ can be selected freely if it is independent of $S$. The posterior $\mathbb{Q}$ can depend on the data. Indeed, a strength of the PAC-Bayesian analysis is that it applies to any $\mathbb{Q}$, and therefore it allows to choose a distribution $\mathbb{Q}$ in a data-dependent manner. The posterior $\mathbb{Q}$ can be optimized to control the divergence between $\mathbb{P}$ and $\mathbb{Q}$. Therefore, we typically have a small KL divergence.

Observe also that, in Theorem 1, when $\mathbb{E}_{\theta \sim \mathbb{P}}[\beta_\theta] \lesssim O(n^{-\frac{1}{2}})$, our generalization bound is $\widetilde{O}(n^{-\frac{1}{2}})$, while (3.3) implies a vacuous bound $O(1)$ so generalization is not guaranteed. Furthermore, our bounds require $\mathbb{E}_{\theta \sim \mathbb{P}}[\beta_\theta] \lesssim 1/\sqrt{n}$ to get almost optimal rates $\widetilde{O}(1/\sqrt{n})$. As a comparison, the results in Eq. (3.3) require stronger condition $\mathbb{E}_{\theta \sim \mathbb{P}}[\beta_\theta] \lesssim 1/n$ and $\beta_\Theta \leq 1/T$ to get the rate $O(1/\sqrt{n})$.

In the unlikely case when the KL divergence is the dominating term, for example, $D_{\mathrm{KL}}(\mathbb{Q}\|\mathbb{P}) \gtrsim O(n)$, then Theorem 2 achieves better result. But in this circumstance, both Theorems 1 and 2 require $\beta_\theta \lesssim O(n^{-\frac{3}{2}})$ to converge at a rate of $O(n^{-\frac{1}{2}})$, which we believe this to be also rather uncommon.

In terms of assumptions, we eliminate the hyperparameter stability assumption in [32] at the price of an additional sub-exponential tail assumption on the uniform stability parameter. We will prove later in Section 4 that this general assumption holds for more stochastic optimization methods and even for non-smooth problems. In conclusion, we achieve better results under weaker conditions.

## 3.2 Proof sketch and challenge in the analysis

We begin by noting that $\ell$ is applied to the output of $A(S;\theta)$, which depends on the training data and the hyperparameters. Therefore, $\ell(A(S;\theta))$ could be sensitive to changes in both the dataset and the hyperparameters. To address this issue, Theorem 2 assumes hyperparameter stability, which requires small changes in $\ell(A(S;\theta))$ when the hyperparameters are perturbed.

In contrast to the previous method, we adopt a different strategy for controlling changes in $\theta$. Our $\beta_\theta$ is a random variable w.r.t. $\theta$. We assume a sub-exponential concentration behavior of $\beta_\theta$ to control its deviation (Assumption 1). This allows us to control $\beta_\theta$ without necessitating hyperparameter stability. Based on this assumption, our proof proceeds by bounding the MGF of the generalization gap. A key challenge is to handle the randomness of $\beta_\theta$. More precisely, for any temporarily fixed $\theta$ it has been shown that $G(S;\theta)$ is a mixture of sub-Gaussian and sub-exponential random variables when only considering the randomness of $S$ [8]. Then a bound of $\mathbb{E}_S[\exp(\lambda G(S;\theta))]$ requires an assumption $\lambda \lesssim 1/\beta_\theta$ (by Eq. (A.5) on sub-exponential random variables). This constraint makes it challenging since we need to choose an appropriate $\lambda$ and control the associated $\mathbb{E}_S\mathbb{E}_{\theta\sim\mathbb{P}}[\exp(\lambda G(S;\theta)]$ (the random constraint makes the selection of $\lambda$ difficult). Our key idea to address this problem is to control the MGF of another function $H : \mathcal{Z}^n \times \Theta \mapsto \mathbb{R}$ defined as follows

$$
H(S;\theta) = \begin{cases} R(A(S;\theta)) - R_S(A(S;\theta)) & \text{if } \theta \in \Omega_\delta, \\ 0 & \text{otherwise,} \end{cases}
$$

where Assumption 1 holds everywhere on $\Omega_\delta$, which is a subset of $\Theta$ with probability measure at least $1 - \delta$. This definition of $H$ has two benefits.

• First, $H(S;\theta)$ is equal to $G(S;\theta)$ with high probability. Therefore, a high-probability bound on $H$ would imply a high-probability bound on $G$.

• Second, $H$ is convenient to control. For any $\theta \in \Theta \setminus \Omega_\delta$, the MGF satisfies $\mathbb{E}_S[\exp(\lambda H(S;\theta)] \leq 1$. For any $\theta \in \Omega_\delta$, the term $\mathbb{E}_S[\exp(\lambda H(S;\theta))]$ can be controlled under an assumption $\lambda \lesssim 1/\mathbb{E}_{\theta\sim\mathbb{P}}[\beta_\theta]$ due to the definition of $\Omega_\delta$ which relates $\beta_\theta$ with its expectation. A key difference is that the random $\beta_\theta$ is replaced by $\mathbb{E}_{\theta\sim\mathbb{P}}[\beta_\theta]$ in the constraint of $\lambda$, which allows us to choose a deterministic $\lambda$ uniformly for all $\theta$. A further expectation w.r.t. $\theta$ would imply a bound on $\mathbb{E}_{S;\theta}[\exp(\lambda H(S;\theta))]$, which is needed in the PAC-Bayes analysis.

Finally, we manipulate this MGF bound in the PAC-Bayes framework by the variational formula (Lemma A.3), building upon the framework of [32].

## 3.3 Bounds in Expectation

Our previous analysis gives high-probability bounds. In this subsection, we give PAC-Bayes bounds in expectation. In this case, we no longer need an assumption on the concentration behavior of $\beta_\theta$ around its expectation. The proof is given in Appendix A.3.

**Theorem 3.** *Consider any $\beta_\theta$-uniformly stable algorithm $A$ and $M$-bounded loss $\ell$. For any distribution $\mathbb{Q}$, we have*

$$
\mathbb{E}_S\mathbb{E}_{\theta\sim\mathbb{Q}}\big[R(A(S;\theta)) - R_S(A(S;\theta))\big] \leq \big(\chi^2(\mathbb{Q}\|\mathbb{P}) + 1\big)^{\frac{1}{2}} \left(\frac{2M^2}{n} + 16\mathbb{E}_{\theta\sim\mathbb{P}}[\beta_\theta^2]\right)^{\frac{1}{2}}. \tag{3.4}
$$

**Remark 1.** Under the same assumption, it was shown in Theorem 1 in [32] that

$$
\mathbb{E}_S\mathbb{E}_{\theta\sim\mathbb{Q}}\big[R(A(S;\theta)) - R_S(A(S;\theta))\big] \leq \big(\chi^2(\mathbb{Q}\|\mathbb{P}) + 1\big)^{\frac{1}{2}} \left(\frac{2M^2}{n} + 12M\mathbb{E}_{\theta\sim\mathbb{P}}[\beta_\theta]\right)^{\frac{1}{2}}. \tag{3.5}
$$

Therefore, our analysis gives a bound of the order $O\big(1/\sqrt{n} + \big(\mathbb{E}_{\theta\sim\mathbb{P}}[\beta_\theta^2]\big)^{\frac{1}{2}}\big)$, while Eq. (3.5) gives a bound $O\big(1/\sqrt{n} + \big(\mathbb{E}_{\theta\sim\mathbb{P}}[\beta_\theta]\big)^{\frac{1}{2}}\big)$. It is known that $\mathbb{E}_{\theta\sim\mathbb{P}}[\beta_\theta^2]$ can be much smaller than $\mathbb{E}_{\theta\sim\mathbb{P}}[\beta_\theta]$. For example, for SGD with $t$ iterations and $\mathbb{P}$ being the uniform distribution, it was shown $\mathbb{E}_{\theta\sim\mathbb{P}}[\beta_\theta] = O(\eta t/n)$ and $\mathbb{E}_{\theta\sim\mathbb{P}}[\beta_\theta^2] = O(\eta^2 t/n)$ if the loss function is convex and smooth [27]. In the typical setting with $\eta = O(1/\sqrt{t})$ and $t \asymp n$ (in this setting SGD achieves optimal rates), we have $\mathbb{E}_{\theta\sim\mathbb{P}}[\beta_\theta] = O(1/\sqrt{n})$ and $\mathbb{E}_{\theta\sim\mathbb{P}}[\beta_\theta^2] = O(1/n)$. In this case, our analysis gives PAC-Bayes bounds of the order $O(1/\sqrt{n})$, while Eq. (3.5) gives PAC-Bayes bounds of the order $O(1/n^{\frac{1}{4}})$. Therefore, our analysis implies much better PAC-Bayes bounds than that in [32].

# 4 Applications

We apply our general results to derive PAC-Bayes bounds for two optimization algorithms: Stochastic Gradient Descent (SGD) and Randomized Coordinate Descent (RCD). To this aim, we introduce some necessary definitions. Let $\|\cdot\|_2$ denote the Euclidean norm.

**Definition 3** (Lipschitz continuity). We say a loss function $\ell(\cdot; z)$ is $L$-Lipschitz if for any $\mathbf{w} \in \mathcal{W}$ and $z \in \mathcal{Z}$, we have $\|\nabla \ell(\mathbf{w}; z)\|_2 \leq L$. This implies that for any $\mathbf{w}_1, \mathbf{w}_2 \in \mathcal{W}$,

$$|\ell(\mathbf{w}_1; z) - \ell(\mathbf{w}_2; z)| \leq L\|\mathbf{w}_1 - \mathbf{w}_2\|_2. \tag{4.1}$$

**Definition 4** (Convexity). Let $\kappa \geq 0$. We say a loss function $\ell(\cdot; z)$ is $\kappa$-strongly convex if for any $\mathbf{w}_1, \mathbf{w}_2 \in \mathcal{W}$ and $z \in \mathcal{Z}$, we have

$$\ell(\mathbf{w}_1; z) \geq \ell(\mathbf{w}_2; z) + \langle \nabla \ell(\mathbf{w}_2; z), \mathbf{w}_1 - \mathbf{w}_2 \rangle + \frac{\kappa}{2}\|\mathbf{w}_1 - \mathbf{w}_2\|_2^2. \tag{4.2}$$

We say the loss function $\ell$ is convex if the above inequality holds with $\kappa = 0$.

**Definition 5** (Smoothness). Let $\alpha \geq 0$. We say a loss function $\ell(\cdot; z)$ is $\alpha$-smooth if for any $\mathbf{w}_1, \mathbf{w}_2 \in \mathcal{W}$ and $z \in \mathcal{Z}$, we have

$$\|\nabla \ell(\mathbf{w}_1; z) - \nabla \ell(\mathbf{w}_2; z)\|_2 \leq \alpha \|\mathbf{w}_1 - \mathbf{w}_2\|_2. \tag{4.3}$$

For the applications, we only need to verify the sub-exponential stability of the algorithm w.r.t. the prior sampling $\mathbb{P}$, which is often chosen to be simple such as the uniform distribution.

## 4.1 Applications to Stochastic Gradient Descent

SGD is one of the most popular algorithms to solve optimization problems in machine learning due to its simplicity and efficiency. The basic idea is to build a stochastic gradient based on a randomly selected example, which is used to update iterates. Here we consider SGD with a general sampling scheme, where the random index follows from a general distribution. This general SGD has already been considered in the literature to improve the efficiency of SGD with uniform sampling, including importance sampling [66] and Markov chain sampling [54, 58, 63].

**Definition 6** (SGD with general sampling). Let $\mathbf{w}_1$ be an initial point. Let $\mathbb{P}$ be a probability measure over $[n]^T$ and $S = \{z_1, \ldots, z_n\}$ be a training dataset. Let $(i_1, \ldots, i_T)$ be drawn according to $\mathbb{P}$. At the $t$-th iteration, SGD with sampling scheme $\mathbb{P}$ updates the model by

$$\mathbf{w}_{t+1} = \mathbf{w}_t - \eta_t \nabla \ell(\mathbf{w}_t; z_{i_t}), \tag{4.4}$$

where $\{\eta_t\}$ is a positive step-size sequence. If $\mathbb{P}$ is the uniform distribution, then we call it SGD with uniform sampling (SGDU).

Now we apply Theorem 1 to develop PAC-Bayes bounds for SGD applied to convex problems, covering both smooth and non-smooth cases. We will show that SGD enjoy sub-exponential stability.

### 4.1.1 Smooth case

In the following lemma to be proved in Appendix II.1.1, we give stability bounds for SGDU and show it satisfies Assumption 1. Recall that the indicator function $\mathbb{I}[\cdot]$ is defined in Definition 2.

**Lemma 4** (Stability bound). *Let $S$ and $S'$ be neighboring datasets. Suppose for all $z \in \mathcal{Z}$ the loss function is convex, $\alpha$-smooth and $L$-Lipschitz. Let $\{\mathbf{w}_t\}, \{\mathbf{w}_t'\}$ be the sequence produced by SGDU on $S$ and $S'$ respectively with $\eta_t \leq 2/\alpha$. Then SGDU with $t$ iterations and the hyperparameter $\theta$ is $\beta_\theta$-uniformly stable with*

$$\beta_\theta = 2L^2 \max_{k \in [n]} \sum_{j=1}^t \eta_j \mathbb{I}[i_j = k].$$

*If $\eta_t = \eta$, then for any $\delta \in (0, 1)$, with probability at least $1 - \delta$ we have*

$$\beta_\theta \leq \mathbb{E}_{\theta \sim \mathbb{P}}[\beta_\theta] + 4L^2 \eta \left(1 + (t/n)^{\frac{1}{2}}\right) \log(1/\delta).$$

*That is, Assumption 1 holds with $c = 4L^2 \eta \left(1 + (t/n)^{\frac{1}{2}}\right)$ w.r.t. $\mathbb{P}$.*

We can combine the above lemma with Theorem 1 to obtain PAC-Bayes bounds for SGD, whose proof is given in Appendix II.1.1. An interesting property is that generalization bounds for SGD with general sampling can be derived based on the stability analysis for SGD with the uniform sampling. We always let $\mathbb{P}$ denote a uniform prior on $\Theta$.

**Corollary 5** (Generalization bound). *Assume $\ell$ is $M$-bounded, $L$-Lipschitz, convex and $\alpha$-smooth. For uniform distribution $\mathbb{P}$, every $n \in \mathbb{N}^+$ and $\delta \in (0,1)$, with probability at least $1 - \delta$ over draws of a data set, $S \sim \mathcal{D}^n$, for all posterior sampling distribution $\mathbb{Q}$ on $[n]^T$, SGD with $\eta \leq 2/\alpha$ satisfies*

$$\mathbb{E}_{\theta \sim \mathbb{Q}}\big[R(A(S;\theta)) - R_S(A(S;\theta))\big] \lesssim$$
$$\Big(D_{\mathrm{KL}}(\mathbb{Q}\|\mathbb{P}) + \log(1/\delta)\Big) \max\Big\{L^2 \eta(T/n + (1 + (T/n)^{\frac{1}{2}})\log(n))\log n, \frac{M}{\sqrt{n}}\Big\}.$$

Based on [20, 27], which studied the trade-off between optimization and stability, the recommended choices of parameters are $T \asymp n$ and $\eta \asymp 1/\sqrt{T}$ to get a SGD iterate with good generalization behavior. In this setting, the above corollary implies a PAC-Bayes bound $\widetilde{O}(1/\sqrt{n})$.

London [32] gave PAC-Bayes bounds for SGD under strong convexity and smoothness assumptions.

**Corollary 6** (Corollary 1 in [32]). *Suppose $\ell$ is $M$-bounded. Let the objective function be $\kappa$-strongly convex, $L$-Lipschitz and $\alpha$-smooth. Then, for uniform distribution $\mathbb{P}$ and any $\delta \in (0,1)$, with probability at least $1 - \delta$ over draws of a data set, $S \sim \mathcal{D}^n$, SGD with $\eta_t = (\kappa t + \alpha)^{-1}$ and any posterior sampling distribution $\mathbb{Q}$ on $[n]^T$ satisfies*

$$\mathbb{E}_{\theta \sim \mathbb{Q}}\big[R(A(S;\theta)) - R_S(A(S;\theta))\big] \lesssim \sqrt{\Big(D_{\mathrm{KL}}(\mathbb{Q}\|\mathbb{P}) + \log(1/\delta)\Big)\Big(\frac{\big(M + L^2/\kappa\big)^2}{n} + \frac{L^2}{\kappa^2 T}\Big)}.$$

**Remark 2.** We now compare Corollary 5 and Corollary 6. First, Corollary 6 requires a strong convexity assumption, which is removed in our analysis. Second, our analysis implies PAC-Bayes bounds of the order $\widetilde{O}(1/\sqrt{n})$, while Corollary 6 implies bounds of order $O(1/(\sqrt{n}\kappa))$. The strong convexity parameter $\kappa$ is often very small in both theoretical and empirical analysis. For example, the existing generalization analysis of regularization schemes suggests $\kappa = O(n^{-\frac{1}{2}})$ to get an optimal bound (Section 3 in [55]), for which Corollary 6 implies a vacuous bound. By contrast, our bound $\widetilde{O}(1/\sqrt{n})$ is optimal up to a log factor.

**Remark 3.** Our uniform stability (Definition 1) is slightly different from the uniform stability $\beta_{\mathcal{Z}} := \sup_{S \sim S'} \sup_z \big|\mathbb{E}_{\theta \sim \mathbb{P}}[\ell(A(S;\theta), z) - \ell(A(S';\theta), z)]\big|$ in [32] in the sense of taking expectation in different places. The expectation of $\theta \sim \mathbb{P}$ is taken outside $\sup$ in our case and inside $\sup$ in $\beta_{\mathcal{Z}}$ [32]. However, we often have similar upper bounds for $\mathbb{E}[\beta_\theta]$ and $\beta_{\mathcal{Z}}$. Consider SGD for smooth, Lipschitz and convex problems as an example. It is shown in [20] that $\beta_{\mathcal{Z}} \leq 2L^2 \sum_{k=1}^t \eta_k/n$, while we can show $\mathbb{E}[\beta_\theta] \lesssim L^2 \log n \sum_{k=1}^t \eta_k/n$ (the proof of Lemma 4). These two upper bounds are the same order up to a logarithmic factor. Furthermore, lower bounds were established in [65] (Theorem 1) where $\beta_{\mathcal{Z}} \geq \frac{L}{2} \sum_{k=1}^t \eta_k/n$, which match the existing upper bounds up to a constant factor. This shows that $\beta_{\mathcal{Z}}$ and $\mathbb{E}[\beta_\theta]$ are of similar order.

### 4.1.2 Non-smooth case

The following lemma shows that SGDU applied to non-smooth problems enjoys the sub-exponential stability. The proof follows the analysis in Section 4.2 in [29] and is given in Appendix II.1.2.

**Lemma 7** (Stability bound). *Let $S$ and $S'$ be neighboring datasets. Suppose for all $z \in \mathcal{Z}$ the loss function is convex and $L$-Lipschitz. Let $\{\mathbf{w}_t\}, \{\mathbf{w}_t'\}$ be the sequence produced by SGDU on $S$ and $S'$ respectively with fixed step sizes. Then SGDU with $t$ iterations and the hyperparameter $\theta$ is $\beta_\theta$-uniformly stable with $\beta_\theta = 2\sqrt{e}L^2 \eta\big(\sqrt{t} + \max_{k \in [n]} \sum_{j=1}^t \mathbb{I}[i_j = k]\big)$. For any $\delta \in (0,1)$, with probability at least $1 - \delta$ we have*

$$\beta_\theta \leq \mathbb{E}_{\theta \sim \mathbb{P}}[\beta_\theta] + 4\sqrt{e}L^2 \eta\big(1 + (t/n)^{\frac{1}{2}}\big)\log(1/\delta).$$

*That is, Assumption 1 holds with $c = 4\sqrt{e}L^2 \eta\big(1 + (t/n)^{\frac{1}{2}}\big)$ w.r.t. $\mathbb{P}$.*

Based on the above lemma, we derive the following corollary for the PAC-Bayes bounds of SGD in non-smooth problems.

**Corollary 8** (Generalization bound). *Assume $\ell$ is $M$-bounded, $L$-Lipschitz and convex. For any $\delta \in (0,1)$, with probability at least $1-\delta$ over draws of a data set, $S \sim \mathcal{D}^n$, for all posterior sampling distribution $\mathbb{Q}$ on $[n]^T$, SGD with $T$ iterations and $\eta_t = \eta$ satisfies*

$$\mathbb{E}_{\theta \sim \mathbb{Q}}\big[R(A(S;\theta)) - R_S(A(S;\theta))\big] \lesssim$$
$$\left(D_{\mathrm{KL}}(\mathbb{Q}\|\mathbb{P}) + \log\frac{1}{\delta}\right) \max\left\{L^2\eta(\sqrt{T} + T/n + (1+(T/n)^{\frac{1}{2}})\log n)\log n, \frac{M}{\sqrt{n}}\right\}.$$

**Remark 4.** If we choose $\eta \asymp T^{-\frac{3}{4}}$ and $T \asymp n^2$, then Corollary 8 gives the PAC-Bayes bounds of order $\widetilde{O}(1/\sqrt{n})$. This choice of parameters was suggested in Theorem 7 in [27]. This gives the $O(1/\sqrt{n})$ optimization bounds to get optimal trade-off between stability and optimization for non-smooth problems. The analysis in [32] cannot imply PAC-Bayes bounds for non-smooth problems.

## 4.2 Applications to Randomized Coordinate Descent

In this subsection, we consider RCD; this has not been studied in the PAC-Bayesian literature. RCD is an efficient optimization algorithm that randomly chooses a coordinate to update at each iteration [37]. Here we consider RCD with general sampling scheme, i.e. the coordinate to update follows a general distribution. This scheme has been studied before in the optimization context [1, 66].

**Definition 7** (RCD with general sampling). Let $\mathbf{w}_1$ be an initial point. Let $\mathbb{P}$ be a probability measure over $[d]^T$ and $S = \{z_1, \ldots, z_n\}$ be a training dataset. Let $(i_1, \ldots, i_T)$ be drawn according to $\mathbb{P}$. At the $t$-th iteration, RCD with sampling scheme $\mathbb{P}$ updates the model by

$$\mathbf{w}_{t+1} = \mathbf{w}_t - \eta_t \nabla_{i_t} R_S(\mathbf{w}_t)\mathbf{e}_{i_t}, \tag{4.5}$$

where $\{\eta_t\}$ is a step-size sequence, $\mathbf{e}_i$ is the $i$-th coordinate vector in $\mathbb{R}^d$, and $\nabla_i g$ is the derivative of $g$ w.r.t. the $i$-th coordinate. If $\mathbb{P}$ is the uniform distribution, then we call it RCD with uniform sampling (RCDU).

Before giving the generalization bound for RCD, we first introduce coordinate-wise smoothness.

**Definition 8** ([37]). A differentiable function $g : \mathcal{W} \to \mathbb{R}$ has coordinate-wise Lipschitz continuous gradients with parameter $\hat{\alpha} > 0$, if for all $\lambda \in \mathbb{R}$, $\mathbf{w} \in \mathcal{W}$, $i \in [d]$,

$$g(\mathbf{w} + \lambda\mathbf{e}_i) \leq g(\mathbf{w}) + \lambda\nabla_i g(\mathbf{w}) + \hat{\alpha}\lambda^2/2.$$

In Lemma 9, to be proved in Appendix B.2, we develop stability bounds for RCDU with convex and smooth loss functions. In particular, we show the stability follows a sub-exponential distribution.

**Lemma 9** (Stability bound). *Let $S$ and $S'$ be neighboring datasets. Suppose for all $z \in \mathcal{Z}$ the loss function $\ell$ is convex, $\alpha$-smooth, $L$-Lipschitz and has coordinate-wise Lipschitz continuous gradients with parameter $\hat{\alpha} \geq 0$. Let $\{\mathbf{w}_t\}, \{\mathbf{w}'_t\}$ be the sequence produced by RCDU on $S$ and $S'$ respectively with $\eta_t \leq 2/\hat{\alpha}$. Then RCD with $t$ iterations is $\beta_\theta$-uniformly stable with*

$$\beta_\theta = \frac{L}{n} \max_{k \in [n]} \sum_{j=1}^t \eta_j |\nabla_{i_j}\ell(\mathbf{w}_j; z_k) - \nabla_{i_j}\ell(\mathbf{w}_j; z'_k)|, \tag{4.6}$$

*where $\|\nabla\ell(\mathbf{w}; z)\|_1 \leq L_1$. Furthermore, if $\eta_t = \eta$, then for any $\delta \in (0,1)$ the following inequality holds with probability at least $1-\delta$ over $\theta \sim \mathbb{P}$ (the uniform distribution over $\{(i_1, \ldots, i_t) : i_j \in [d]\}$)*

$$\beta_\theta \leq \frac{2L_1 L\eta t}{nd} + \frac{\eta L^2 \log(1/\delta)}{n}\left(\frac{8}{3} + \sqrt{\frac{32t}{d}}\right).$$

**Remark 5.** Stability bounds of the order $O(\eta t/(nd))$ were developed for RCD in [62] (Theorem 2). Their stability bounds hold in expectation. In contrast, we develop high-probability stability bounds of the order $O\left(\frac{\eta t}{nd} + \frac{\eta}{n}\right)$.

We plug the above bounds into Theorem 1, and derive the following PAC-Bayes bounds for RCD.

**Corollary 10** (Generalization bound). *Let the assumptions in Corollary 5 hold. We further assume that the gradient is coordinate-wise Lipschitz continuous. When $\mathbb{P}$ is the uniform distribution, for any $\delta \in (0,1)$, with probability at least $1-\delta$ over draws of $S \sim \mathcal{D}^n$, for all posterior sampling distributions $\mathbb{Q}$ on $[d]^T$, RCD with the hyperparameter $\theta$ and fixed step sizes $\eta \leq 2/\hat{\alpha}$ satisfies*

$$\mathbb{E}_{\theta \sim \mathbb{Q}}\big[R(A(S;\theta)) - R_S(A(S;\theta))\big] \lesssim \left(D_{\mathrm{KL}}(\mathbb{Q}\|\mathbb{P}) + \log(1/\delta)\right) \max\left\{\frac{LL_1\eta T}{nd}\log n, \frac{M}{\sqrt{n}}\right\}.$$

According to the above corollary, we can derive PAC-Bayes bounds of the order $\widetilde{O}(1/\sqrt{n})$ if $T = O(d\sqrt{n})$ and $\eta = O(1)$. These choices of parameters were suggested in Theorem 7 in [62] to balance optimization and stability for RCD.

## 5  Related Work

**Related work on stability**    The analysis of the generalization error through algorithmic stability is based on the landmark work of [7]. Generalization bounds via stability are algorithm-specific and have been applied to regularization algorithms, such as SVM regression and classification [7, 11, 16, 52]. Pioneering work on stability analysis of SGD was introduced in [20], which motivated much subsequent stability analyses of randomized iterative algorithms [10, 24, 25, 31, 38, 45]. The smoothness assumption in [20] was removed in more recent stability analyses [3, 27] of SGD. Stability bounds showing the benefit of low training errors on generalization were also developed for convex [27, 39, 50], nonconvex [28] and overparameterized models [12, 47, 59]. In recent works, stronger high-probability bounds via uniform stability have been developed. Some breakthroughs have narrowed down the difference between the risk and the empirical risk, leading to faster convergence rates with high probability [8, 17, 18].

**Related work on PAC-Bayes bounds**    The PAC-Bayes theory of generalization dates back to works by [53] and [34] and further improved by [9, 26] and others. PAC-Bayes bounds are upper bounds on the generalization error of randomized learning algorithms, given in terms of data-dependent quantities that we can compute: the empirical error, and a quantity to measure the divergence between the PAC-Bayesian prior and posterior distributions, such as Kullback-Leibler (KL) divergence or the Rényi divergence [4]. The framework was later extended to allow learning the prior from the data, resulting in tighter bounds [2, 14, 15, 35, 42–44, 48]. The sensitivity of learning algorithms to small perturbations in the weights can be analyzed through the properties of a distribution of predictors, which may lead to regularity and improve the generalization bounds [5]. By evaluating the algorithmic stability on all possible outputs, stability of learning algorithms and PAC-Bayes bounds can be combined [32, 33, 35, 36, 40, 48, 57] and applied to randomized learning algorithms such as SGD and SGLD [30, 32, 35, 36]. For PAC-Bayes bounds, usually, the randomness is induced by a distribution on the parameters of a model. In [32], the authors isolated the randomness to view the randomized learning algorithm as deterministic, with hyperparameters following the distribution instead.

## 6  Conclusions

Under an assumption of a sub-exponential stability parameter, we derive sharper stability-based PAC-Bayes bounds for randomized learning algorithms by utilizing a moment bound. We show that the sub-exponential stability assumption holds for SGD and RCD, for which we develop PAC-Bayes bounds as corollaries. Our results remove the need for the requirements of strong convexity, hyperparameter stability, and even smoothness in previous results.

**Limitations**. Future work of interest includes exploring other optimization methods that meet the sub-exponential assumption. It would also be interesting to study the quality of bounds obtainable when placing the PAC-Bayes prior on the parameters of a model (as in the classic approach) versus the hyperparametrs of the optimiser of the model (as in this work). Another interesting research direction is to further improve the bound by including additional assumptions. As noted in [23], for the deterministic case, the bounds can be $\sqrt{n}$-times faster under a Bernstein condition between expectation and variance of loss functions.

**Acknowledgements**. The authors are grateful to the anonymous reviewers for their thoughtful comments and constructive suggestions. The work of Yunwen Lei is partially supported by the Research Grants Council of Hong Kong [Project No. 22303723]. The work of Sijia Zhou is funded by CSC and UoB scholarship. AK acknowledges funding by EPSRC Fellowship grant EP/P004245/1.

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
