# OpenReview forum: "Toward Better PAC-Bayes Bounds for Uniformly Stable Algorithms"
_NeurIPS.cc/2023/Conference — NeurIPS 2023 poster_

### Official Review · Reviewer_qzEi · 2023-07-04

**Soundness:** 3 good
**Presentation:** 3 good
**Contribution:** 3 good
**Rating:** 7
**Confidence:** 2

**Summary:**

This paper gave an improved PAC-Bayes bound for uniformly stable randomized algorithms. It introduced new assumptions and proof techniques. The paper also applied the general theorem to concrete applications (SGD and RCD) and got better or novel PAC-Bayes bound for these two algorithms.

**Strengths:**

1. The general PAC-Bayes bound improvement in this paper is good. Although the improvement doesn't always hold, it is sharper in most cases and can be up to $\sqrt{n}$, which is significant. This result seems to be quite non-trivial.
2. The authors introduced new assumptions and proof techniques. The newly introduced assumption of sub-exponential stability is interesting and provided a new perspective compared to the previous assumption. The new proof technique is innovative and could be studied and used on proving other problems.
3. The paper gave concrete applications to the general theorem, which shows the power of such bound. The bound for SGD is sharper and requires no strong convexity and the bound for RCD is completely novel. These results are nice contributions.
4. The paper is well-written. As someone who doesn't know much about PAC-Bayes, reading the paper doesn't take too much effort. Sufficient background information is given, definitions are clearly and maths are rigorous. Contributions and novelty of the paper is highlighted properly.

**Weaknesses:**

1. One thing I find confusing is that the prior and posterior are over the hyperparameters but not on the hypothesis class, while usually in the PAC-Bayes framework the prior and the posterior are defined over the hypothesis class. This part should be clarified, I think more background or explanation is needed.
2. The use of the word "parameter" and "hyperparameter" is a bit confusing because the meanings seem to be different from its usual meaning. Actually the parameter class $W$ barely shows in the paper and I think it is basically equivalent to $H$ here. From my understanding, "hyperparameter" here is just the random seed of the algorithm, but usually it means something like the step size. Maybe changing the names could make it better.
3. A very tiny typo: line 101 should be "parameters" instead of "parameter".

**Questions:**

1. In line 79, it says "given the parameter $w\in W$", what does this mean? Shouldn't the algorithm output $w$ instead of given $w$? Or do you mean "given the parameter class $W$"?
2. In line 183-185 you claimed that the benefit of PAC-Bayes bound is that the prior and posterior can be selected and posterior parameters can be optimized? I understand the prior can be selected freely but isn't the posterior depends on $S$ and the algorithm? This part may need more explanation.
3. If my understanding is correct, in both of your applications (SGD and RCD), even in the generalized version, the sampling distribution is still a fixed distribution across all iterations. Can you extend your result to algorithms where the sampling distribution could be different in every iteration or even depend on the history?


**Limitations:**

The authors addressed the limitation of their paper properly.

---

> ### Author Rebuttal · Authors · 2023-08-09
>
> Thank you for your constructive comments and suggestions.
>
>   **Q1**.  The prior and posterior are over the hyperparameters but not on the hypothesis class.
>
>    **A:** Thanks for the suggestion. We build on the idea of [29] to consider the prior and posterior on the hyperparameters instead of that on the hypothesis class. In the context of randomized algorithms, it is more natural to handle the randomness through the distribution on the hyperparameters as compared to hypothesis. Then, we can learn a good posterior distribution and draw a hyperparameter from this posterior distribution for the implementation of an algorithm. Another benefit is that the posterior distribution on the hyperparameters is usually simple, while the associated distribution on hypothesis can be very complicated. For example, the paper [R1] designs an importance sampling strategy for SGD, where at each iteration the index is drawn from a non-uniform distribution on $\{1,2,\ldots,n\}$. While this posterior distribution on hyperparameters is simple, the resulting distribution of models produced by SGD can be very complicated. We will give more background in the revised version.
>
>   **Q2**.  The use of the word "parameter" and "hyperparameter" is a bit confusing.
>
>    **A:** Thanks for indicating this. We agree that this may cause confusion, and will clarify as follows. As in [29] we use the term ''hyperparameters'' to mean random seeds. In particular, we will use "weights" to mean $\mathbf{w}$. We will emphasize that  hyperparameters mean the random seeds of the algorithm in our setting, as in [29].
>
>   **Q3**.  What is the meaning of "given the parameter $\mathbf{w}\in W$"?
>
>    **A:** Thanks for indicating this. We wanted to mean $h_{\mathbf{w}}$ is determined by $\mathbf{w}\in W$. We will modify it as ''maps the training examples to a hypothesis $h_{\mathbf{w}}\in\mathcal{H}$ determined by $\mathbf{w}\in W$''.
>
>   **Q4**.  I understand the prior can be selected freely but isn't the posterior depends on data and the algorithm? This part may need more explanation.
>
>    **A:** Thanks for the suggestion. Yes, the prior can be selected freely if it is independent of the data $S$. The posterior can depend on the data $S$. Indeed, a strength of the PAC-Bayesian analysis is that it applies to any posterior distribution. The posterior distribution can even depend on the data, and therefore it allows to choose a distribution $\mathbb{Q}$ in a data-dependent manner. For example, [R2,R3] and [29] trained data-dependent posterior distributions to minimize the bounds. We will add more explanations in the revised version.
>
>
>   **Q5**.  For both SGD and RCD, the sampling distribution is still a fixed distribution across all iterations. Can you extend your result to algorithms where the sampling distribution could be different in every iteration or even depend on the history?
>
>    **A:** Yes, while we only consider the uniform distribution in checking the sub-exponential condition for SGD and RCD, our results apply to any posterior distributions. The sampling distribution of the algorithm can be different in very iteration or even depend on the histroy. Indeed, the posterior distribution $\mathbb{Q}$ is the distribution of the indices $\theta=(i_1,\ldots,i_T)$, where $i_t$ is drawn from a sampling distribution at the $t$th iteration. The only requirement is to compute the KL divergence between $\mathbb{p}$ and $\mathbb{Q}$. A strength of the combining stability and PAC-Bayesian together is that it only needs to verify the stability for the prior distribution $\mathbb{P}$. Indeed, we only need to study stability for the prior distribution, and then use the idea of PAC-Bayesian analysis to transfer this result to any posterior distribution. Therefore, we choose the simple uniform distribution as the prior and check whether SGD and RCD satisfy Assumption 1 (measured under the prior distribution). We will clarify this in the revised version.
>
>
> [R1] P. Zhao and T. Zhang. Stochastic Optimization with Importance Sampling, ICML, 2015.
>
> [R2] M. Perez-Ortiz, O. Rivasplata, J. Shawe-Taylor, and C. Szepesvari. Tighter risk certificates for neural networks, Journal of Machine Learning Research, 2021.
>
> [R3] G. K. Dziugaite and D. M. Roy. Computing nonvacuous generalization bounds for deep (stochastic) neural networks with many more parameters than training data, arXiv preprint arXiv:1703.11008, 2017.

---

> > ### Comment · Reviewer_qzEi · 2023-08-15
> >
> > Thanks for your response. I will keep my score.

---

### Official Review · Reviewer_qrJQ · 2023-07-06

**Soundness:** 1 poor
**Presentation:** 2 fair
**Contribution:** 2 fair
**Rating:** 4
**Confidence:** 4

**Summary:**

The work aims to provide stronger generalization guarantees for stochastic algorithms through the analysis of stability and PAC-Bayes. Such algorithms include SGD and RCD. The work uses a milder assumption in the analysis under the same problem setting as [29]. The results can be extended beyond strong convexity from previous results and hold for non-smooth convex problems.

**Strengths:**

If the results are correct, they are tighter than a previous work [29] in the same problem setting. Also, they require milder assumptions (uniform stability and sub-exp stability), whereas [29] requires some stability w.r.t. the changes of the data and the hyperparameters. Therefore, although they build from the setting in [29], they require different analysis techniques.

**Weaknesses:**

The significance of the results is not very clear. Please see more details in Question 1.

The soundness of the results is not clear. Please see more details in the next section from point 2 and beyond.

**Questions:**

1. Although it’s a similar setting as [29], it’s unclear why you need PAC-Bayes analysis to guarantee all distributions Q over the hyperparameter space. It doesn’t seem like you are trying to learn some good distributions Q of randomized rule. As far as I can see, you are interested in providing a generalization guarantee for some fixed distribution, regardless of how arbitrary it is. Therefore, isn’t it enough to have results for a fixed distribution like in [15]?

2. I feel there is something missing in the results related to assumption 1. It looks like assumption 1 is a characterization of a randomized algorithm A defined by some distribution P. And it essentially says all deterministic algorithms defined by \theta in the support of P are nice in the sense of being \beta_\theta-uniform stable. Also, the distribution P itself is nice in the sense that the \beta_\theta from such P is w.h.p. not too far from the mean with some parameter \tilde b.

a) Is the stability constant \tilde b in assumption 1 a function of the distribution P?

b) It’s perhaps better to say, in line 136, that it’s uniform stable “w.r.t. a loss function \ell” since it makes little sense to talk about stability without mentioning the loss. Also, the \ell\in[0,M] assumption in line 152 should perhaps be associated with the stability conditions in assumption 1 rather than be put independently.

c) Again, assumption 1 is a characterization of some A defined by some P. Hence, it’s not clear what it means by “a learning algorithm A that satisfies Assumption 1” (in line 152) without mentioning the associated distribution. Also, it’s most likely that other distributions Q won’t follow the same sub-exp stability constant \tilde b as P (the formula in theorem 1).

3. Do you have any idea why Theorem 1 doesn’t depend on the loss range M? It is neither in the bound nor in the stability constant \beta_\theta in your example in Lemma 3.

4. In Lemma 3, the SGDU is explicitly associated with a distribution P=uniform. It’s not clear to me why this randomized algorithm is \beta_\theta uniformly stable, which only depends on some specific \theta.

**Limitations:**

There were quite some fast-rate results in PAC-Bayes analysis in other applications. There are also some fast-rate analyses for the algorithms addressed in the work but with stronger assumptions on losses. However, due to the “max” function in theorem 1, the results here cannot go beyond O(1/sqrt(n)) convergence rate. Whether a fast rate is possible here can be an interesting question.

---

> ### Author Rebuttal · Authors · 2023-08-09
>
> Thank you for your constructive comments and suggestions.
>
>   **Q1**. It is unclear why you need PAC-Bayes analysis to guarantee all distributions $\mathbb{Q}$. Isn't it enough to consider a fixed distribution like in [15]?
>
>    **A:** Thank you for the intuitive question. Since the PAC-Bayes bound holds uniformly for any posterior $\mathbb{Q}$, we are allowed to select the best $\mathbb{Q}$ that minimises (or approximately minimises) the bound, despite it depends on the training set. In other words we can learn $\mathbb{Q}$ from the data by minimising the bound, i.e. select a distribution $\mathbb{Q}$ that balances the trade off between minimizing empirical error and reducing the KL divergence between the prior and posterior. Indeed [29] demonstrates an algorithm that learns $\mathbb{Q}$ by approximately minimising the PAC-Bayes bound. The algorithm associated with our work is the same as there, our contribution is an improved analysis / convergence rate. Namely, it suffices to study uniform stability under the prior distribution and we have the flexibility to choose any posterior distribution. This can not be achieved in [15], as that requires the choice of $\mathbb{Q}$ to be made before seeing the data.
>
>   **Q2**.  I feel something missing in results related to Assumption 1, which is a characterization of $A$ and says all deterministic algorithms are $\beta_\theta$-uniformly stable. Also, $\beta_\theta$ is not far from the mean.
>
>  **A:** Thank you for the comment. In Assumption 1 we consider a randomized algorithm $A$ with a random hyperparameter $\theta\sim\mathbb{P}$. For each $\theta$, we have a deterministic algorithm. For $\theta \sim \mathbb{P}$ we can estimate the uniform stability $\beta_\theta$ of the algorithm defined in Definition 1, which is a function of random $\theta$. In Assumption 1, we assume $\beta_\theta$ concentrates around its expectation w.h.p., under the distribution $\mathbb{P}$. Take SGD with $T$ iterations as an example. Then we have $\theta=(i_1,\ldots,i_T)$ and we derive an upper bound of $\beta_\theta$ as a function of random $(i_1,\ldots,i_T)$. With concentration inequality, we then show that $\beta_\theta\leq \mathbb{E}[\beta_\theta]+O(\eta(1+t/n))$ w.h.p. in Lemma 3, where the expectation is $\theta\sim\mathbb{P}$. A crucial point is that in our PAC-Bayesian analysis, we only need to check the stability for any fixed $\mathbb{P}$. Therefore, we can choose $\mathbb{P}$ to be simple, e.g., the uniform distribution over $\{(i_1,\ldots,i_T)\}$ in the case of SGD. We will add more discussions in the revision.
>
>   **Q3**.  Is the stability constant $\tilde b$ a function of $\mathbb{P}$?
>
>    **A:** Yes. Since the assumption is made w.r.t. $\mathbb{P}$, the term $\tilde{b}$ depends on $\mathbb{P}$. For SGD, we give an explicit form of $\tilde{b}$ in Lemma 3 when $\mathbb{P}$ is a uniform distribution.
>
>   **Q4**.  It makes little sense to say stability without mentioning loss. $\ell\in[0,M]$ should be associated with Assumption 1.
>
>    **A:** Yes, the stability depends on the loss function. To avoid confusion and for brevity, we will emphasize that we always mean stability w.r.t. $\ell$ when mentioning stability. We do not associate $\ell\in[0,M]$ with Assumption 1 since stability may not depend on $M$. For example, in Lemma 3 we show Assumption 1 holds without dependency on $M$, which is consistent with existing stability bounds [19].
>
>   **Q5**.  "a learning algorithm that satisfies Assumption 1" is not clear without mentioning distribution. Other distributions $\mathbb{Q}$ won't follow the same sub-exp stability.
>
>    **A:** Thanks for the comment. When we mention Assumption 1 we always mean Assumption w.r.t. $\mathbb{P}$. The underlying reason is that our analysis only requires Assumption 1 to hold for any fixed $\mathbb{P}$. We do not need to check Assumption 1 for the posterior distribution $\mathbb{Q}$. Indeed, we use PAC-Bayesian analysis to transfer the stability for the particular $\mathbb{P}$ to get results holding for any posterior distribution $\mathbb{Q}$. To avoid confusion, we will emphasize in the revision that we always mean the prior distribution $\mathbb{P}$ when mentioning Assumption 1.
>
>   **Q6**. Do you have any idea why Theorem 1 doesn't depend on the loss range $M$? It is neither in the bound nor in the stability constant $\beta_\theta$ in your example in Lemma 3.
>
>    **A:** Theorem 1 actually depends on $M$. We showed the dependency on $M$ in the proof (Line 550), and absorbed it in the $\lesssim$ notation in Theorem 1. We will state them clearly in the revision. $\beta_\theta$ may not depend on $M$. Instead, it depends on $L$ [19], and we include this dependency in Lemma 3.
>
>   **Q7**.  In Lemma 3, the SGDU is explicitly associated with a distribution $\mathbb{P}=\text{uniform}$. It is not clear to me why this randomized algorithm is $\beta_\theta$ uniformly stable.
>
>    **A:** Thank you for the comment. We give proof for the uniform stability of SGDU in both smooth and nonsmooth cases from line 568 to line 585 and from line 590 to line 605. A strength of our result is that we only need to check the sub-exponential condition of the stability for any prior distribution $\mathbb{P}$. Then we use PAC-Bayesian analysis to transfer the stability for the particular $\mathbb{P}$ to get results holding for any posterior distribution $\mathbb{Q}$. For SGD with $T$ iterations, the randomness comes from the sampling of index $\theta=(i_1,\ldots,i_T)$. For each $\theta$, we get a deterministic algorithm. Therefore, for $\theta \sim \mathbb{P}$, we can estimate the uniform stability by using the convexity, Lispchitz and smoothness assumption, following the idea of existing stability analysis of SGD [19]. This stability bound for SGD with the hyperparameter $\theta$ depends on $\theta=(i_1,\ldots,i_T)$. Since $\theta$ is a random hyperparameter, we further use concentration inequality to show that $\beta_\theta$ is a sub-exponential random variable.

---

> > ### Comment · Reviewer_qrJQ · 2023-08-15
> > **Follow-up**
> >
> > Thank you for the clarification of the questions.
> >
> > Regarding Q1, I understand that [29] considered this already, and I understand having a guarantee for all Q allows you to choose from (data-dependent) distributions. I might be missing something, but it doesn’t seem needed in the user cases you demonstrate. To justify it is useful, it will be helpful to see a case where such data-dependent Q can be efficiently obtained, and it really will be better than a uniform distribution.
> >
> > Regarding Q7, my question was unrelated to the translation between P and Q. SGDU is associated with a distribution P, which is a distribution over \theta. I was confused by why SGDU is \beta_\theta uniformly stable, which is a property of some \theta from P. It looks like using a property of an instance to describe the underlying distribution.
> >
> > For Q2-Q6, I will highly recommend making the dependency more explicit.  And it will perhaps be helpful to emphasize you are/and only have to look at a chosen P.

---

> > > ### Author Response · Authors · 2023-08-17
> > > **Reply to Follow-up**
> > >
> > > Thank you for the further queries.
> > >
> > > **Q: It will be helpful to see a case where such data-dependent $\mathbb{Q}$ can be efficiently obtained, and it really will be better than a uniform distribution.**
> > >
> > > **A**. An example is the importance sampling in [R1], where SGD selects a random index from a non-uniform distribution according to the importance of training examples. In particular, [R1] proposes to draw $i_t\in\{1,2,\ldots,n\}$ based on the probability $\mathbf{q}=(q_1^t,q_2^t,\ldots,q_n^t)$, where
> > > $$
> > > q_i^t\propto ||\nabla \ell (w_t;z_i)||.
> > > $$
> > > Experimental results in [R1] show that SGD with this importance sampling scheme achieves improved performance as compared to SGD with the uniform sampling.
> > >
> > > Another example is the adaptive sampling algorithm for SGD that optimizes the posterior at runtime [29]. As motivated by the PAC-Bayesian analysis, this algorithm selects $i_t$ according to (proportionally) $\mathbf{q}=(q_1,\ldots,q_n)$ and $q_{i_t}$ is updated as
> > > $$q_{i_t}=q_{i_t}^\tau\exp(\alpha f(w_t,z_{i_t})),$$
> > > where $\tau\in(0,1)$ is a decay parameter, $\alpha\geq0$ is an amplitude parameter and $f(w_t,z_{i_t})$ is a utility function of $w_t$ on the chosen example $z_{i_t}$. Experimental results in [29] show that adaptive sampling can reduce empirical risk faster than uniform sampling while also improving out-out-sample accuracy.
> > >
> > > We will add more discussions in the revision to clarify the motivation. However the conceptual novelty and contribution of our work is the theoretical analysis.
> > >
> > >
> > > **Q: Regarding Q7, I was confused by why SGDU is $\beta_\theta$ uniformly stable.**
> > >
> > > **A**: Sorry for causing confusion. We wanted to mean that first we condition on $\theta$ -- that is, SGDU with a *given* hyperparameter $\theta$ (i.e., $\theta$ has already been drawn from $\mathbb{P}$ and given) is $\beta_\theta$-uniformly stable, cf. our Definition 1. Indeed, a basic idea in checking the sub-exponential condition (our Assumption 1) is to first estimate $\beta_\theta$ for any given $\theta$. Then we use the fact that $\theta$ follows from the uniform distribution to show that $\beta_\theta$ is a sub-exponential random variable. We will make this clear in the statement of Lemma 3.
> > >
> > > **Q: For Q2-Q6, I will highly recommend making the dependency more explicit. And it will perhaps be helpful to emphasize you are/and only have to look at a chosen $\mathbb{P}$.**
> > >
> > > **A**: We agree. In the revision, we will make the dependency more explicit and emphasize the sufficiency of considering a single $\mathbb{P}$.
> > >
> > > [R1] P. Zhao and T. Zhang. Stochastic Optimization with Importance Sampling, ICML, 2015.

---

### Official Review · Reviewer_V6pn · 2023-07-07

**Soundness:** 3 good
**Presentation:** 3 good
**Contribution:** 3 good
**Rating:** 7
**Confidence:** 3

**Summary:**

In this paper, the authors study PAC-Bayesian bounds on the generalization error of uniformly stable algorithms. They consider algorithms in which the randomness comes from a particular hyperparameter $\theta$, e.g., stochastic gradient descent (SGD), in which the randomness comes from selecting the indices used to determine which samples are utilized in the gradients. This way, they consider algorithms that are stable for every choice of the hyperparamter $\theta$, but they allow different degrees of stability $\beta\_\theta$ for every parameter. They only require that the stability parameter is at least exponentially close to its mean under some prior distribution $\mathbb{P}$, that is $\beta\_\theta \leq \mathbb{E}\_{\vartheta \sim \mathbb{P}} [\beta\_\vartheta] + \tilde{b} \log(1/\delta)$ with probability $1 - \beta$ over the draws of $\theta$ (sub-exponential condition).

In this setup, they obtain generic PAC Bayes bounds that scale linearly with the KL divergence between a posterior over the parameters $\mathbb{Q}$ and the prior $\mathbb{P}$. These bounds have the same rate as the bounds from Bousquet, Klochkov, and Zhivotovskiy [8, Corollary 8] when the parameters are deterministic. Moreover, compared to the PAC-Bayes bounds from London [29, Theorem 2], they substitute the parameter uniform stability by the sub-exponential condition, which is weaker, and their bound still has a better rate.

In terms of applications:
* They show that SGD with a Lipschitz, convex, and smooth loss is stable for all parameters $\theta$ selecting the indices to determine which samples are utilized in the gradients, and that that the parameters satisfy the sub-exponential condition for a uniform prior distribution. This way, they can obtain a generalization bound with rate $\tilde{O}(1/\sqrt{n})$, improving upon London's bounds [9] that require strong convexity and potentially have a worse rate.
* They show that the smoothness assumption can be lifted and there are still choices of the learning rate and number of iterations under which the rate is $\tilde{O}(1/\sqrt{n})$.
* They show that randomized coordinate descent (RCD) with a Lipschitz, convex, smooth, and coordinate-Lipschitz loss is also stable for all parameters $\theta$ used to select the coordinate where the gradient is taken, and  that the parameters satisfy the sub-exponential condition for a uniform prior distribution. This way, they obtain bounds of a similar order to Wang, Wu, and Lei [56] that hold with high probability instead than in expectation.

**Strengths:**

In terms of originality, the main source of originality is the consideration of per hyper-parameter uniform stability + a sub-exponential condition in the stability parameter. The analysis employed for finding the bound for a fixed hyper-parameter is very similar to Bousquet, Klochkov, and Zhivotovskiy [8], but the consideration of a sub-exponential condition on the stability parameter and the introduction of said condition to find generic bounds is novel.

Similarly, the proofs to show that SGD and RCD are stable with a sub-exponential stability parameter are based on common analyses in the field + standard concentration arguments. However, combining them into this setting is also new as far as I know.

In general, the idea of introducing a prior and a posterior over the parameters did not seem very relevant in the beginning, as many times the hyper-parameters distribution is independent of the data and one can always have $\mathbb{Q} = \mathbb{P}$ with a zero divergence. However, it appeared to be quite clever afterwards since one only needs to prove the sub-exponential condition for the prior and see how far the actual distribution is from that.

The results seem significant since the control of the sampling is often a drawback in many analyses of SGD and its variants, that have very specific forms of sampling. Also, some of the results obtained are already as good as previous results in the literature when reduced to their setting, or more general, or better.

**Weaknesses:**

I believe that some things could be clarified. For instance what I mentioned in the strengths about the introduction of a prior and a posterior. Some commentary about that would be helpful. At least for readers like me, who may be wondering why would one introduce this seemingly artificial distribution over the hyper-parameters.

Also, it would be good to clarify that in Corollaries 4,7, and 9 the prior $\mathbb{P}$ must be the uniform distribution. Similarly, it would be easier for the reader if in Definition 7 you would use the same letter used fro priors throughout, i.e. $\mathbb{P}$.

Sometimes in the text and in the appendices, references are cited without the specific place where one should look. This difficults the reader experience. For instance, For Lemma A.2 only [55] is mentioned, when it refers to a combination of Propositions 2.5.2 and 2.7.1. I hope the authors go back to their text and look at their citations and try to specify better where the reader should look.

In Lemma A.1, the authors seem to use the version v1 of the paper [8] in the arxiv. They may want to look at Theorem 4 in the latest version (and the version actually published). They will see that the constants are improved: $3 \sqrt{2}$ is actually $4$ and $\sqrt{6}$ is actually $\sqrt{2}$.

Lemma A.3 is not the Donsker-Varadhan formula. The Donsker-Varadhan formula deals with the supremum of the functions $h$ appearing in their equation, finding a dual equivalence. The shown theorem is the Gibbs variational principle (see e.g. Lemma 4.10 of [A]).

There seem to be some mistakes in the constants in the proof of Theorem 1. Not very important as they seem to be solvable:
* After line 538 shouldn't it be 36 instead of 72?
* After line 538, where did the multiplicative factor of 2 in the second exponential?
* It seems you forgot a + 3 in the right hand side of equation 547. This should also appear in the general statement of the theorem as $(D\_{\mathrm{KL}}(\mathbb{Q} \Vert \mathbb{P}) + \log \frac{1}{\delta} + 3)$. Shouldn't it?

**Additional References**

Ramon van Handel. "Probability in High Dimension". APC 550 Lecture Notes. Princeton University. 2016.

**Questions:**

* I don't understand how you go from the equation after line 593 to the equation after line 540.
* Could you elaborate on more cases in Remark 2? From what I see, in Remark 2 using the standard $T \in \Theta(n)$ and a small learning rate $\eta \in O(1/\sqrt{n})$ yields a reasonable rate of $\tilde{O}(1/\sqrt{n})$ as well.

Repeating here some of the questions in the weaknesses:
* After line 538 shouldn't it be 36 instead of 72?
* After line 538, where did the multiplicative factor of 2 in the second exponential?
* It seems you forgot a + 3 on the right-hand side of equation 547. This should also appear in the general statement of the theorem as $(D\_{\mathrm{KL}}(\mathbb{Q} \Vert \mathbb{P}) + \log \frac{1}{\delta} + 3)$. Shouldn't it?

**Limitations:**

The authors discuss some of the limitations throughout the text and in the conclusion.

---

> ### Author Rebuttal · Authors · 2023-08-09
>
> Thank you for your constructive comments and suggestions.
>
>   **Q1**.  It would be better to clarify why would one introduce this seemingly artificial distribution over the hyper-parameters.
>
>    **A:** Thanks for the suggestion. We will add more discussions on the distribution of the hyper-parameters. In particular, we will emphasize that it allows us to build a PAC-Bayesian bound holding for any arbitrary posterior distribution, which allows us to learn a posterior distribution to minimize the bound and draw a hyperparameter from this posterior distribution for the implementation of the algorithm.
>
>   **Q2**.  It would be good to clarify that in Corollaries 4,7, and 9 the prior must be the uniform distribution.
>
>    **A:** We agree, and will follow your suggestion to mention it clearly in the revision.
>
>   **Q3**.  Sometimes in the text and in the appendices, references are cited without the specific place where one should look. For instance, For Lemma A.2 only [55] is mentioned, when it refers to a combination of Propositions 2.5.2 and 2.7.1.
>
>    **A:** Thank you for indicating this and checking the reference. Yes, Lemma A.2 is a combination of Propositions 2.5.2 and 2.7.1 in [55].  We will mention the specific result when citing a result in a book in the revision.
>
>   **Q4**.  In Lemma A.1, the authors seem to use the version v1 of the paper [8] in the arxiv.
>
>    **A:** Yes, you are right. We will use the improved constants in the new version. Thanks for indicating this.
>
>   **Q5**. Lemma A.3 is not the Donsker-Varadhan formula. The shown theorem is the Gibbs variational principle (see e.g. Lemma 4.10 of [A]).
>
>    **A:** Thank you for indicating this. We will modify it and update the reference in the revision.
>
>   **Q6**. I don't understand how you go from the equation after line 539 to the equation after line 540.
>
>    **A:** Thank you for checking the details. The r.h.s. of the inequality after line 539 is of the form $\exp(a) \cdot (\exp(b) +\exp(c)) +\delta$, where $a,b,c>0$ are some terms indicated in the inequality.
>
>   Then, we actually use the following inequality to get the equation after line 540
>
>   $$\exp(a) \cdot (\exp(b) +\exp(c)) +\delta\leq \exp(a+1/2) \cdot (\exp(b) +\exp(c)).$$
>
>   This is equivalent to
>
>   $$\delta \leq (\exp(b) +\exp(c))\exp(a)\big(\sqrt{e}-1\big).$$
>
>   This holds directly since $\delta<1,\exp(b)+\exp(c)\geq2$ and $\sqrt{e}-1\geq1/2$. We will give more details in the revision.
>
>  **Q7**.  Could you elaborate on more cases in Remark 2?
>
>    **A:** Thank you for the comment. If we choose $T=\Theta(n)$, then the bound would be of the order $O(\eta\sqrt{n}
>    +1/\sqrt{n})$ (ignoring KL divergence and log factors). Then $\eta$ should be of order $O(1/n)$ to get the generalization bound $O(1/\sqrt{n})$. In this case, the $\eta=O(1/T)$ would be too small to get a good decay of optimization error.
>
>  **Q8**.  Line 538: coefficient should be $36$ instead of $72$?
>
>    **A:** Thanks for checking the details. Yes, it should be $36$ instead of $72$. We will modify it.
>
>   **Q9**. Line 538, where did the multiplicative factor of 2 in the second exponential?
>
>    **A:** Thanks for the careful reading. We want to bound $\mathbb{E}\exp(2\lambda X_2)$ and therefore should apply Eq (A.12) with $\lambda$ replaced by $2\lambda$.
>
>   **Q10**. Line 547, it seems you forgot $a + 3$ in the right hand side of equation 547.
>
>    **A:** Thank you for careful reading. Yes, there should have $+3$ in the proof and we absorbed it in the big $O$ notation in the statement of Theorem 1. We will modify it in the revised version.

---

> > ### Comment · Reviewer_V6pn · 2023-08-10
> > **Answer to rebuttal**
> >
> > Dear Authors,
> >
> > Thank you for your answers and for incorporating most of my comments into the paper.
> >
> > Sincerely,
> >
> > Reviewer V6pn

---

### Official Review · Reviewer_ZHyT · 2023-08-02

**Soundness:** 3 good
**Presentation:** 3 good
**Contribution:** 2 fair
**Rating:** 5
**Confidence:** 3

**Summary:**

In this paper, the authors extend the work of London [29] and prove PAC-Bayesian generalization bounds for uniformly stable algorithms. Compared to [29], the present work removes some required assumptions in the work of [29], such as the hyperparameter stability. To do so, they required a new assumption called "Sub-exponential stability" and proved Theorem 1 based on this new assumption. They further instantiate their theorem for stochastic gradient descent and randomized coordinate descent.

**Strengths:**

- The hyperparameter stability assumption is not required in the present analysis.
- The strong convexity assumption in the SGD applications is removed compared to the analysis of [29].
- To the best of my knowledge, this is the first time that Randomized Coordinate Descent is studied in the literature.

**Weaknesses:**

- I find the $\lesssim$ and the $\widetilde{O}$ notations sometimes confusing (see the comments below).
- The statements of the theorems of [29] do not seem to be correctly restated.
- There are no experiment evaluations.


**Questions:**

_Major comments_:

- **l173**: In the current form of Theorem 2, I could not compare it with Theorem 2 of [29]. Indeed, for me, in [29], their bound includes the term $\beta_{\mathcal{Z}}$ while you have instead $\mathbb{E}\_{\theta\sim\mathbb{P}}[\beta\_\theta]$. Indeed, by looking at the two definitions, i.e. your Definition 1 and one in [29], the expectation over $\mathbb{P}$ is not in the same place.
- **l179-181**: I am confused with your $\widetilde{O}(\cdot)$ notation and your discussion below.
Indeed, your discussion involves the KL divergence, but it is hidden in the two equations. Hence, the discussion is hard to understand in this current form.
- **l270**: In [29], $R\_S$ does not have to be necessarily $\gamma$-strongly convex and $\alpha$-smooth. Indeed, they consider another function (called $F$ in their notations) to be minimized.

_Minor comments_:

- **l152**: I would have expected to have $M$ in the bound since you introduced the notation in the statement of the theorem.
- **l494**: You cite the wrong Theorem in [8]; you cite an old version of the paper. By the way, the constants are improved in the new version of the theorem.
- **l512**: Since [55] is a book, can you cite the section or the theorem of the Lemma for readability?

_Typos_:

- **l154**: There is ".." at the end of the sentence.
- **In Appendix B**: there is some "$-$" that are in fact "$+$".

**Limitations:**

I do not see any potential negative societal impact of their work since it is theoretical.

---

> ### Author Rebuttal · Authors · 2023-08-09
>
> Thank you for your constructive comments and suggestions.
>
> **Q1**. L173: In your Definition 1 and one in [29], the expectation over $\mathbb{p}$ is not in the same place.
>
> **A:** Thanks for the careful reading and indicating this. $\mathbb{E}[\beta_\theta]$ in our submission is different from $\beta_{\mathcal{Z}}$ in [29] in the sense of taking expectation in different places: expectation is outside $\sup$ in $\mathbb{E}[\beta_\theta]$, and inside $\sup$ in $\beta_{\mathcal{Z}}$. However, we often have similar upper bounds for $\beta_{\mathcal{Z}}$ and $\mathbb{E}[\beta_\theta]$. Consider SGD for smooth, Lipschitz and convex problems as an example. The paper [19] shows $\beta_{\mathcal{Z}}\leq 2L^2\sum_{k=1}^t\eta_k/n$, while we show $\mathbb{E}[\beta_\theta]\leq 2L^2\sum_{k=1}^t\eta_k/n$ in the proof of Lemma 3. These two upper bounds are the same. Furthermore, lower bounds were established in [R1, Thm 1]: $\beta_{\mathcal{Z}}\geq \frac{L}{2}\sum_{k=1}^t\eta_k/n$, which match the existing upper bounds up to a constant factor. This shows that $\beta_{\mathcal{Z}}$ and $\mathbb{E}[\beta_\theta]$ are of the same order.  Also, the analysis in [29] involves $\beta_{\Theta}$ which takes supremum over $\theta,z,S$. We remove this term by introducing the sub-exponential condition. Here the expectation is taken with respect to $\theta\sim\mathbb{P}$ in $\mathbb{E}[\beta_\theta]$.
>
> **Q2**. L179-181: I am confused with your $\widetilde{O}$ notation and your discussion below. Indeed, your discussion involves the KL divergence, but it is hidden in the two equations. Hence, the discussion is hard to understand in this current form.
>
> **A:** Thanks for indicating this. We will add the term $D_{\text{KL}}(\mathbb{Q}\|\mathbb{P})$ in the equations and revise it as follows: we show
>
>    $$\mathbb{E}\_{\theta\sim\mathbb{Q}} \big [R(A(S;\theta))-R_S(A(S;\theta))\big] =  \widetilde{O}\big(\mathbb{E}\_{\theta\sim\mathbb{P}}[\beta_\theta]+
>    n^{-\frac{1}{2}}\big)D_{\text{KL}}(\mathbb{Q}\|\mathbb{P}),$$
>
>   while the analysis in [29] shows
>
>   $$\mathbb{E}\_{\theta\sim\mathbb{Q}} \big [R(A(S;\theta))-R_S(A(S;\theta))\big] = \widetilde{O}\big(\sqrt{n}\mathbb{E}\_{\theta\sim\mathbb{P}}[\beta_\theta] + \sqrt{T} \beta_\Theta\big)D^{\frac{1}{2}}_{\text{KL}}(\mathbb{Q}\|\mathbb{P}).$$
>
>   Then, if the KL divergence is dominated by the stability parameter, our bound can be sharper up to a factor of $\sqrt{n}$.
>
>   **Q3**.  L270: In [29], $R_S$ does not have to be necessarily strongly convex and smooth. Indeed, they consider another function (called $F$ in their notations) to be minimized.
>
>    **A:** The analysis in [29] requires the objective function $F$ to be strongly convex, Lipschitz and smooth, which, as you indicated, may not be $R_S$. As a comparison, we do not require the objective function to be strongly convex, and therefore our analysis can apply to SGD to minimize $R_S$ directly. We will clarify this confusion by modifying the statement of Corollary 5 (Corollary 1 in [29]).
>
> **Q4**. L152: I would have expected to have $M$ in the bound since you introduced the notation in the statement of the theorem.
>
> **A:**  Yes, the bound depends on $M$. We showed the dependency on $M$ in the proof of Theorem 1 (Appendix A.1, Line 550), and absorbed it in the $\lesssim$ notation in the statement of Theorem 1. We will state them clearly in Theorem 1.
>
>   **Q5**.  L494: You cite an old version of the paper [8].
>
>  **A:** Thanks for indicating this. We will cite the new version and modify the constants.
>
>  **Q6**.  L512: Can you cite the section or the theorem of the Lemma for readability?
>
>  **A:** Thank you for the suggestion. We agree. We refer to Prop. 2.5.2 and Prop. 2.7.1 in [55]. We will mention it clearly in the revision.
>
>  **Q7**. Some Typos.
>
>  **A:** Thank you for spotting the typos. We will fix them in the revision.
>
> [R1]: Stability of SGD: Tightness Analysis and Improved Bounds. Zhang et al. UAI, 2022.

---

> > ### Comment · Reviewer_ZHyT · 2023-08-15
> >
> > Thank you for your answer. I raised the score by one point.

---

### Decision · Program_Chairs · 2023-09-21

**Decision:**

Accept (poster)

**Comment:**

The work is a rigorous contribution to the PAC-Bayesian analysis of randomized learning algorithms. The clever introduction of a "sub-exponential assumption on the stability parameter" allows for an improvement in the previous result of London (2017). Other researchers might subsequently build on this rigorous analysis to further improve theoretical guarantees and "self-certified" learning algorithms.

The reviewers ask for many clarifications and provide suggestions during the discussion period. It is important that the authors incorporate these in the camera-ready of the paper.